# Bottom-Up Strategy to Forecast the Drug Location and Release Kinetics in Antitumoral Electrospun Drug Delivery Systems

**DOI:** 10.3390/ijms24021507

**Published:** 2023-01-12

**Authors:** Raffaele Longo, Marialuigia Raimondo, Luigi Vertuccio, Maria Camilla Ciardulli, Marco Sirignano, Annaluisa Mariconda, Giovanna Della Porta, Liberata Guadagno

**Affiliations:** 1Department of Industrial Engineering, University of Salerno, Via Giovanni Paolo II 132, 84084 Fisciano, Italy; 2Department of Engineering, University of Campania “Luigi Vanvitelli”, Via Roma 29, 813031 Aversa, Italy; 3Department of Medicine, Surgery and Dentistry, University of Salerno, Via S. Allende, 84081 Baronissi, Italy; 4Department of Chemistry and Biology, University of Salerno, Via Giovanni Paolo II, 132, 84084 Fisciano, Italy; 5Department of Science, University of Basilicata, Viale dell’Ateneo Lucano 10, 85100 Potenza, Italy; 6Interdepartment Centre BIONAM, Università di Salerno, Via Giovanni Paolo I, 84084 Fisciano, Italy

**Keywords:** electrospinning, melanoma topical treatment, drug release, matrix-filler affinity

## Abstract

Electrospun systems are becoming promising devices usable for topical treatments. They are eligible to deliver different therapies, from anti-inflammatory to antitumoral. In the current research, polycaprolactone electrospun membranes loaded with synthetic and commercial antitumoral active substances were produced, underlining how the matrix-filler affinity is a crucial parameter for designing drug delivery devices. Nanofibrous membranes loaded with different percentages of Dacarbazine (the drug of choice for melanoma) and a synthetic derivative of Dacarbazine were produced and compared to membranes loaded with AuM1, a highly active Au-complex with low affinity to the matrix. AFM morphologies showed that the surface profile of nanofibers loaded with affine substances is similar to one of the unloaded systems, thanks to the nature of the matrix-filler interaction. FTIR analyses proved the efficacy of the interaction between the amidic group of the Dacarbazine and the polycaprolactone. In AuM1-loaded membranes, because of the weak matrix-filler interaction, the complex is mainly aggregated in nanometric domains on the nanofiber surface, which manifests a nanometric roughness. Consequently, the release profiles follow a Fickian behavior for the Dacarbazine-based systems, whereas a two-step with a highly prominent burst effect was observed for AuM1 systems. The performed antitumoral tests evidence the high-cytotoxic activity of the electrospun systems against melanoma cell lines, proving that the synthetic substances are more active than the commercial dacarbazine.

## 1. Introduction

The research for innovative active substances and cutting-edge drug delivery systems are undoubtedly among the most relevant challenges in the biomedical field. However, with the current scientific progress that rapidly overcomes barriers with more effective therapies and more accurate diagnoses, it is crucial to understand the general chemical-physical principles that shape the behavior of the systems to easily forecast how the synthesis of more effective active substances, the development of more performing materials [1] and the progress in more advanced processes can be judiciously exploited and applied based on precedent researches. Generally, it is known that matrix-drug affinity is a relevant parameter that can sensitively affect drug encapsulation and its release [2,3,4,5,6]. For example, in liposomes, hydrophilic drugs are generally encapsulated in the core, whereas lipophilic drugs are encapsulated in the external phospholipid bilayer membrane [7]. However, these types of systems (microcapsules, liposomes, etc.), which are undoubtedly applicable for certain delivery forms (e.g., injection), cannot always be suitable for topical treatments. For example, after surgery, the patient may require combined treatments, from wound dressing to therapeutic delivery [8]. In this scenario, it is evident that oncological surgery requires devices that can provide the active substance to the patient to efficiently enhance wound healing. Electrospinning is surely one of the processes that meet these requirements since it allows producing highly flexible membranes loaded with active fillers. Furthermore, the obtained nanofibrous morphology guarantees good skin breathing in the zone of interest [9] and simultaneously mimics the features of the extracellular matrix, reducing the healing time for the wound [10]. For this reason, this technology seems ideal for solid and superficial tumors, such as melanomas, one of the leading causes of death from skin cancer [11]. This field of study attracts the scientific community’s interest, considering that the therapies against malignant melanoma still lack efficacy due to the heterogeneity [12] and metastatic dissemination [13] of this cancer kind [14,15]. Different types of fillers have been successfully loaded in electrospun membranes [16,17], but understanding how to tune the drug release is still unclear.

Among techniques used to control drug release, one of the most promising is coaxial electrospinning since it allows to obtain of efficient bilayer nanofibers of different polymers [18]. The coaxial electrospinning generally aims to control the active substance release by adding the shell layer, which limits the burst effect from the membrane and allows a slower release in time. However, it also presents some disadvantages, especially in the control of the process that involves a large number of parameters. The coaxial electrospinning generally aims to control the active substance release through the addition of the shell layer, which limits the burst effect from the membrane and allows a slower release in time. However, it also presents some disadvantages, especially in the control of the process that involves a large number of parameters.

Coaxial electrospinning, which shows great potentialities for obtaining tuneable release curves [19] is still not fully exploited because it is not supported by a thorough understanding of how to predict drug release in the uniaxial nanofiber.

The current research paper studies how to forecast the main properties of the drug delivery device, from the release profile to the morphology, based on the supramolecular interactions between the matrix and the filler. Hence, the study aims to investigate the applicability of these systems in the melanoma field by loading uniaxial electrospun membranes with synthetic and commercial antitumoral agents.

Steffens et al. [20] recently evaluated the inclusion of dacarbazine in electrospun membranes of polyvinyl-alcohol, evaluating the antitumoral activity of the obtained systems. To the authors’ knowledge, here electrospun membranes of Polycaprolactone (PCL) loaded with different percentages of dacarbazine, a drug of choice against melanoma [21], have been produced for the first time and tested against human malignant melanoma derived from lymph node metastatic site (MeWo cell line). Similarly, PCL membranes loaded with synthesis gold complexes (AuM1) have been produced and compared to the ones loaded with dacarbazine.

The two compounds present several differences, and the intermolecular interactions have been investigated to determine how they affect the morphology and the release kinetic of the system. In light of relevant results from recent literature [22,23,24] and based on highly promising antitumoral results of AuM1 [25], a derivative of the Dacarbazine has been synthesized and complexed with Au (named (NHC^d^)AuCl), to evaluate the potentiality of a possibly enhanced antitumoral activity and simultaneously to design membranes with release profiles and morphological features sensitively different to the ones loaded with AuM1. The chemical structure of the active substances used is reported in Figure 1.

## 2. Results and Discussion

### 2.1. Chemistry

AuM1 was synthesized according to the procedure already reported in the literature [25].

#### Synthesis of (NHC^d^)AuCl Complex

The synthesis scheme of the proligand (A) and (NHC^d^)AuCl complex (B) is shown in Figure 2.

Synthesis of proligand: N, N′ dimethyl-4-[(E)-dimethylaminodiazenyl]-5-carboxamide imidazolium iodide was obtained by reaction of 4-[(E)-dimethylaminodiazenyl]-2H-imidazole-5-carboxamide (Dacarbazine) with iodomethane (CH_3_I). The product was characterized by ^1^H and ^13^C NMR and MALDI (see Appendix A), and it was obtained with a good yield (80%). Synthesis of gold complex (NHC^d^)AuCl:N, N′ dimethyl-4-[(E)-dimethylaminodiazenyl]-5-carboxamide imidazolyden gold (I) chloride was synthesized by reaction of the proligand with gold (I)-chloro-(dimethylsulfide) [(Me_2_S)AuCl] in inert nitrogen atmosphere. The corresponding (NHC^d^)AuCl complex was obtained in 30% yield (see Figure 2B) [26,27,28,29,30,31,32]. The new complex was characterized by ^1^H and ^13^C NMR, mass spectroscopy, and elemental analysis. ^1^H and ^13^C NMR spectra show the expected signals (the attributions are reported in the Appendix A, see Appendix A) and the sharp resonance for carbene was at 178.43 ppm. The elemental analysis gives a ratio among ligand, gold and chloride of 1:1:1 and mass spectrometry analysis allows to establish the structure of the gold complex. Indeed, MALDI-MS spectrum (see Appendix A) show the peak leading, associated with the compound at 617.22056 m/z, corresponding to [(NHC)_2_Au]^+^. Thus, the complex is constituted of [(NHC^d^)_2_Au]^+^cation and of [AuCl_2_]^−^ anion as it was been shown for an analogous silver complex by the solid-state structure determined by X-ray diffraction [33]. It is worth note that, from literature, a dynamic equilibrium in solution is known between mono and biscarbenic metallic species. The ^1^H and ^13^C NMR and MALDI mass spectra are all reported in the Appendix A.

### 2.2. FTIR Analysis

FTIR spectra of PCL, 3%AuM1, 3%Dacar, and 3%(NHC^d^)AuCl were detected to determine the presence of interactions between the active substance and the hosting polymeric matrix. The typical peaks of the PCL remain unchanged in the FTIR spectra, probably because of the small amount of filler loading in the nanofibers. Figure 3 shows the spectrum of the PCL membrane.

However, by focusing on the 3000–3500 cm^−1^ wavenumber range (Figure 4), it is possible to track down one of the typical peaks of the dacarbazine due to the N-H stretching of the amide group of the dacarbazine. Interestingly, the same peak is evident also in the 3%Dacarbazine membrane, shifting from 3383 cm^−1^ to 3372 cm^−1^. The NH_2_ group of the dacarbazine gives hydrogen bonding with the carbonyl group of the PCL and, for this reason, the N-H bondings are affected [34] by the intermolecular interactions with the matrix. A bathochromic shift to lower wavenumber (so higher wavelength and, so, lower energy) is observed when dacarbazine is included in PCL.

The significant affinity between the PCL matrix and the dacarbazine (and (NHC^d^)AuCl) rather than AuM1, which has been proved via FTIR analysis, strongly influences the placement of the active substance along the nanofibers during the electrospinning process, as widely explored in the AFM analysis section.

### 2.3. AFM Analysis

AFM analysis has been performed to define the nanofiber size in the electrospun membranes and to study the surface characteristics of the nanofibrous membranes. From precedent research of the group [21], the peculiar morphology of electrospun membranes can be investigated via AFM to determine where the filler is loaded during the process. As reported in Figure 5, even if for all the produced membranes the morphology is nanofibrous, several peculiarities are evident for the different matrix-drug systems. For PCL membranes (in the first row of Figure 5), PCL is electrospun without the presence of fillers, and from the AFM morphology, it is possible to notice that the PCL electrospun membranes are very smooth. This information is confirmed by the roughness parameters (R_a_ and R_q_) and phase acquisitions. Similar results, both in height and phase images, have also been obtained for the membranes loaded with Dacarbazine (1% and 3%), with roughness parameters that are very similar to the ones of PCL. In the case of membranes loaded with 1% and 3% AuM1, the feature of the superficial morphology is sensitively different. The AuM1 complex, which has a low affinity with the PCL matrix, tends to be segregated in a large amount on the surface of the nanofibers. Natu et al. [35] reported that when the filler is loaded in a fibrous delivery system above the solubility threshold, it tends to be segregated on the fiber surface. However, given that the solubility depends strictly on the affinity between the matrix and the filler, it is possible to forecast where the active substance tends to be distributed, just considering the interactions between the matrix and the filler. In this case, the intermolecular interactions (hydrogen bonding type) between PCL and Dacarbazine, confirmed via FTIR, allow the inclusion of the Dacarbazine homogeneously in the nanofiber. The absence of strong intermolecular interactions between AuM1 and PCL causes the placement of AuM1 away from the PCL core and, therefore, its segregation on the nanofiber surface. (NHC^d^)AuCl gives interactions with PCL likewise Dacarbazine-PCL membranes, being so included in all PCL material of the fibers. Hence, no segregation effect of the active substance is observed on the fiber surface, as evidenced by the last row of Figure 5.

In Table 1, the roughness parameters evaluated for the unloaded membrane and the membranes loaded with AuM1, Dacarbazine, and (NHC^d^)AuCl are reported. By comparing the results, it is evident that the roughness sensitively increases as low as the affinity between the filler and matrix. In fact, by comparing the results obtained for the membranes loaded with Dacarbazine and the unloaded membranes, the roughness parameters are very similar, confirming the inclusion of the active substance in the polymeric matrix.

### 2.4. Active Substance Release

The encapsulation efficiency of the active substance in the different membranes has been evaluated, and the results are reported in Table 2. The results highlight that the encapsulation efficiency is almost total (corresponding to the theoretical value) for the membranes at 1% of loading, whereas it decreases to 75% for the membranes loaded at 3% of the active substance. The “theoretical value” is the total amount of complex solubilized in the solution before the electrospinning process.

The release curves were evaluated for the membranes loaded with the various active substances via UV-Vis spectroscopy. In order to evaluate the drug release of the drug encapsulated in the membranes, the active substance released has been defined as reported in Equation (1).
(1)$$Active\;Substance\;Released\;\left[\%\right]=\frac{m\left(t\right)}{m_{theoretical\;complex}*\frac{\eta }{100}}*100$$

The release curves display several differences both in terms of the mechanism of release and in terms of kinetic. As expected, 1%AuM1 and 3%AuM1 have a relevant burst effect, given the presence of the complex on the nanofiber surface. As soon as the membranes come into contact with the PBS solution, the crystallite aggregates on the nanofibers start to dissolve in the solution. Only when the complex on the surface has dissolved does the diffusive contribution, due to the complex inclusion in the nanofiber, start to manifest. However, this contribution is limited since most of the active substance was segregated on the nanofiber surface because of the little interaction with the matrix. By contrast, 1%Dacar, 3%Dacar, 1%(NHC^d^)AuCl, and 3%(NHC^d^)AuCl, in which the active substance is included inside the nanofiber structure, have significantly slower release curves, as reported in Figure 6.

The different release mechanisms are confirmed by the first principles modeling analysis. Firstly, a Fickian diffusive release has been hypothesized for all the systems, obtaining the diffusivity term by modeling the curves in the first region of the curves (around 1 h) according to the approximation in Equation (2).
(2)$$\frac{M_{t}}{M_{\infty }}<0.6;\;\frac{M_{t}}{M_{\infty }}=4*{\left(D*\frac{t}{\pi *L^{2}}\right)}^{\frac{3}{2}}$$

Then, the fitting diffusivity parameter has been applied to all the systems using the generalized Fick model for these types of systems [36]. The applied Fick model is reported below in Equation (3)
(3)$$\frac{M_{t}}{M_{\infty }}=1-\frac{8}{\pi^{2}}*\overset{\infty }{\underset{n=0}{\sum }}\frac{\exp\left(-D*{\left(2n+1\right)}^{2}*\pi^{2}*\frac{t}{L^{2}}\right)}{{\left(2n+1\right)}^{2}}$$

However, as it is evident in Figure 6, the Fickian model does not describe accurately 1%AuM1 and 3%AuM1 releases. In fact, the diffusive parameter (D), evaluated in the first region of the sample, is affected by the burst effect, which is sensitively lower/absent for 1%Dacar, 3%Dacar, 1%(NHC^d^)AuCl, and 3%(NHC^d^)AuCl. In fact, the diffusivity for all these last systems is around 10^−8^ mm^2^/s, whereas is 100 times higher (around 10^−6^ mm^2^/s) for AuM1-membranes. This is because the Fick model does not describe appropriately these types of systems. Semi-empirical models derived from the Weibull model, here reported in Equation (4), recently applied in literature [37,38] are more efficient to describe when two mechanisms of release occur.
(4)$$\frac{M_{t}}{M_{\infty }}=\theta *[1-\exp\left(-\frac{1}{A_{1}}*t^{b_{1}}\right)]+\left(1-\theta \right)*[1-\exp\left(-\frac{1}{A_{2}}*t^{b_{2}}\right)]$$

It is worth noticing that the asymptotic value for the membranes loaded with Dacarbazine, which have a higher affinity to the matrix, is generally lower than the one obtained for AuM1-loaded matrices. It is probably due to the higher matrix-filler interaction that leads to holding a higher quantity in the polymer matrix [39].

By comparing the very first zone of the curve, the difference is even more evident. In the first region for 3%(NHC^d^)AuCl and 3%AuM1, is evident how changing the ligand of the complex is possible to obtain completely different release profiles. The results are reported in Figure 7.

In Table 3 are reported the Weibull Parameters evaluated for AuM1 loaded membranes. In Table 4 below, the Fickian diffusive parameters for the membranes are reported.

### 2.5. Cytotoxicity of Free Compounds and PCL Functionalized Membranes

Compared to the commercial Dacarbazine, the synthetic complexes AuM1 and Au-Dacarbazine showed better performance against MeWo. The concentrations of 5 μM and 50 μM were identified as the Minimum Inhibitory Concentrations (MIC) for AuM1 and Au-Dacarbazine, respectively (Figure 8).

Additionally, by testing PCL functionalized membranes, the best antitumoral activity was observed for the synthetic systems. In particular, the synthetic complexes had again a significantly higher activity than the commercial Dacarbazine, showing a cell viability of less than 50% at 24 h and 48 h, as reported in Figure 8. MeWo cell line has been chosen because of its well-known chemo-resistance, but the current research stresses the possibility to approach this challenge in two ways: firstly, the possibility to synthesize completely new metal-complex chemotherapeutics (e.g., AuM1) with high effectiveness against the specific cell line; secondly, the possibility to integrate the proven activity of the metal developing cutting-edge metal complexes, starting from the proved knowledge of the ligand.

## 3. Materials and Methods

### 3.1. Materials

Reagents were purchased from TCI Chemicals and used as received. NMR spectra were recorded at room temperature on a Bruker AVANCE 400 spectrometer (400 MHz for ^1^H; 100 MHz for ^13^C). NMR samples were prepared by dissolving about 8 mg of compounds in 0.4 mL of deuterated solvent. The ^1^H NMR and ^13^C NMR chemical shifts are referenced to the residual proton impurities of the deuterated solvents with respect to SiMe_4_ (*δ* = 0 ppm) as internal standards singlet signals, which are abbreviated as (s). Elemental analyses for C, H, and N were obtained by a Thermo-Finnigan Flash EA 1112 according to standard microanalytical procedures. Chloride was determined by precipitation of AgCl, using AgNO_3_ as a reagent. The silver content was determined after dissolution in Na_2_S_2_O_3,_ by flame atomic absorption spectroscopy (FAAS), and the halogen content was calculated by using the content of silver. MALDI-MS: mass spectra were obtained by a BrukerSolariX XR Fourier transform ion cyclotron resonance mass spectrometer (BrukerDaltonik GmbH, Bremen, Germany) equipped with a 7 T refrigerated actively-shielded superconducting magnet (BrukerBiospin, Wissembourg, France). The samples were ionized in positive ion mode using the MALDI ion source (BrukerDaltonik GmbH, Bremen, Germany). The mass range was set to m/z 200–3000. The laser power was 28%, and 22 laser shots were used for each scan.

Perstorp purchased Poly(-caprolactone) (PCL molecular weight of 80,000 Da; CAS number 24980-41-4). (Molecular weight 80,000 Da). Acetone and Dimethylformamide (DMF-CAS 68-12-2) were bought from Carlo Erba (Cornaredo, Italy). Oxoid (Basingstoke, UK) was purchased as was phosphate-buffered saline (PBS pH 7.3).

### 3.2. Synthesis of N, N′ Dimethyl-4-[(E)-dimethylaminodiazenyl]-5-carboxamide Imidazolium Iodide

To begin, 4-[(E)-dimethylaminodiazenyl]-2H-imidazole-5-carboxamide (Dacarbazine 1.00 mmol, 0.1820 g) and potassium carbonate (K_2_CO_3_, 2.00 mmol, 0.2760 g) were suspended in acetone (25 mL) and stirred at room temperature for 2 h in a nitrogen atmosphere. Then, iodomethane was added to the reaction mixture (CH_3_I, 10.0 mmol, 1.412 g, 0.60 mL), and the solution was stirred for another 24 h at refluxing temperature. The product was recovered by filtration, and the solvent was removed at reduced pressure and washed with hexane (3 × 20 mL) and diethyl ether (2 × 20 mL), obtaining the N, N’ dimethyl-4-[(E)-dimethylaminodiazenyl]-5-carboxamide imidazolium iodide (0.800 mmol, 0.2701 g). Yield: 80%.

### 3.3. Synthesis of N, N′ Dimethyl-4-[(E)-dimethylaminodiazenyl]-5-carboxamide Imidazolyden Gold(I) Chloride—(NHC^d^)AuCl

The imidazolium salt (0.500 mmol, 0.1690 g), chloro(dimethylsulfide) gold(I) (Au(SMe_2_)Cl, 0.500 mmol, 0.1470 g) was suspended in anhydrous ethanol in a round bottom flask. The reaction mixture was stirred for 3 h at room temperature. Then, potassium carbonate (7.50 mmol, 1.036 g) was added, and the mixture was stirred for another 6 h at room temperature. The gold complex (0.150 mmol, 0.0660 g) was obtained by filtration of the reaction mixture and removing the solvent in vacuo. Yield: 30%

The structure of the complex was determined by ^1^H- and ^13^C-NMR, MALDI mass spectrometry, and elemental analysis. ^1^H COSY and NOESY experiments allowed to assign all proton resonances of the ^1^H NMR spectrum, whereas resonances of ^13^C NMR spectrum were attributed using the support of DEPT-135 experiment. All signal attributions are reported in the experimental part.

For (NHC^d^)AuCl complex, the elemental analysis provides the predictable composition in C, H, N, with a ratio of NHC:Au:Cl of 1:1:1. Mass spectroscopy analysis gives only a peak at 617.22056 Da corresponding to [(NHC^d^)_2_Au]^+^. These data have allowed us to determine the true structure of the complex: [(NHC^d^)_2_Au]^+^[AuCl_2_]^−^. The spectra are reported in the Appendix A.

### 3.4. Electrospinning Procedure

PCL pellets have been put into a 3:1 volume mixture of acetone and DMF at a weight of 11%. In the various solutions, complexes have been added in varying weights: 0%, 1%, and 3% with respect to the polymer amount. To obtain homogeneous solutions, the solutions were maintained under magnetic stirring at 40 °C for 24 h.

The climate-controlled electrospinning apparatus (EC-CLI by IME Technologies, Spaarpot 147, 5667 KV, Geldrop, The Netherlands) was used to electrospun the polymeric solutions. An illustrative scheme of the electrospinning process is reported in Figure 9.

Each solution was placed into a 5 mL syringe and fed via a 0.8 mm needle linked to a power supply. In presence of a strong electric field that enables the polymer to spin, the flow was ejected from the needle. The process parameters have been set to reach a steady spinnability of the solution. For all the membranes, the climate chamber conditions were established at 25 °C and 35% relative humidity. The additional process parameters are reported in Table 5.

### 3.5. Sample Preparation and Sterilization Protocol

AuM1, Dacarbazine, and (NHC^d^)AuCl (free compounds) were solubilized in dimethyl sulfoxide (DMSO) and diluted in DMEM high glucose (Gibco^TM^, Waltham, MA, USA) at a final concentration of 1, 5, 10, 20, 50, 100, 200, 300 and 400 μM, for cell treatments.

For adhesion culture, not functionalized and functionalized PCL membranes were cut to obtain a cycle shape of 15 mm of diameter and then placed deep in 70% ethanol and washed twice in sterile 1X PBS (Corning Cellgro, Manassas, VA, USA). Samples were dried for 24 h under a laminar flow cabinet. Silicon rings were cut using a hollow cutter (outer diameter: 14 mm; inner diameter: 11 mm) and sterilized in 70% ethanol. After ethanol evaporation, silicon rings were stuck on PCL membranes using a non-corrosive silicon rubber and let to dry overnight. Samples were deep in 70% ethanol, exposed to UV rays for 5 min on both sides, and then used for cell seeding.

### 3.6. Cell Culture

To test the cytotoxicity of free compounds, human melanoma cells (MeWo; ATCC^®^, HTB-65TM) (P22) were seeded in 96-well plates at a density of 100.000 cells/mL. Cells were cultured in DMEM high glucose (Gibco^TM^, Waltham, MA, USA) containing 10% Fetal Bovine Serum (Gibco^TM^, Waltham, MA, USA), 100 ng/mL streptomycin, 1% Penicillin/Streptomycin (Corning, Manassas, VA, USA) and 1% Glutagro^TM^ (Corning, Manassas, VA, USA) at 37 °C in a 5% CO2 atmosphere. After 24 h, different concentrations of each compound (1, 5, 10, 20, 50, 100, 200, 300, 400 μM) were added to respective wells and incubated for 24 h and 48 h.

To test the cytotoxicity of functionalized PCL membranes, human melanoma cells (MeWo; ATCC^®^, HTB-65^TM^) (P27) were seeded within silicon rings to prevent cells from flushing from the membrane area at a density of 30.000 cells/cm^2^. Samples were cultured in DMEM high glucose (Gibco^TM^, Waltham, MA, USA) containing 10% Fetal Bovine Serum, 10% Fetal Bovine Serum (Gibco^TM^, Waltham, MA, USA), 100 ng/mL streptomycin, 1% Penicillin/Streptomycin (Corn-ing, Manassas, VA, USA) and 1% Glutagro^TM^ (Corning, Manassas, VA, USA) at 37 °C in a 5% CO_2_ atmosphere.

### 3.7. Cell Viability Assay

After the treatment, 0.5 mg/mL of 3-(4,5-Dimethylthiazol-2-yl)-2,5-diphenyl-tetrazolium bromide (MTT) was directly added in the culture medium. MTT solution was incubated for 4 h at 37 °C, protecting the plate from the light. Then, the supernatant was removed, and 100 μL of 100% DMSO was added to solubilize formazan crystals.

The absorbance was measured at 490 nm using a microplate reader (Infinite F200 PRO, Tecan Group Ltd., SW, Mannedorf, Switzerland). Cell metabolic activity was calculated as a percentage compared to the control group (considered as 100%), according to the Equation (5).
(5)$$\%\;\mathrm{Cell}\;\mathrm{viability}=\frac{Absofsample-Absblank}{Absofcontrol-Absblank}\times 100$$

### 3.8. Statistical Analysis

Results from multiple experiments (*n* = 3) are presented as mean  ±  standard deviation (SD). Statistical analysis was performed using an ordinary one-way analysis of variance (ANOVA) test for independent groups. *p* values less than 0.05 were accepted as significant [40]. All statistical analysis was conducted using GraphPad Prism software (6.0 for Windows, LLC, San Diego, CA, USA).

### 3.9. Fourier Transform Infrared Spectroscopy

Fourier Transform Infrared Spectroscopy (FTIR) was carried out using a Bruker Vertex 70 FTIR-spectrophotometer (Bruker Optics Inc., Billerica, MA, USA). The analysis was performed in the range 4000–360 cm^−1^, with a resolution of 4 cm^−1^, and 16 scans were collected. The FTIR on all the samples was performed in absorbance on a thin film of the membranes (≈40 μm).Before each analysis, the membranes were kept under vacuum for 24 h (at room temperature to not compromise the morphology) to avoid absorbed humidity. The infrared spectra on Dacarbazine, AuM1, and (NHC^d^)AuCl were obtained in absorbance using KBr pellets as supports.

### 3.10. Morphology Analysis

The morphology of the materials was examined using atomic force microscopy (AFM). By utilizing the interactions between the tip and the sample surface, AFM is a technique that can recreate a tridimensional image of the surface of the sample. The data are obtained by measuring the cantilever deflection. BrukerNanoScope V multimode AFM (Digital Instruments, Santa Barbara, CA, USA) has been used in ambient atmosphere and ambient atmosphere. The tip used has a nominal spring constant of 20–100 N/m, resonance frequencies of 200–400 kHz, and a tip radius of 5–10 nm. The utilized tip has a tip radius of 5–10 nm, a nominal spring constant of 20–100 N/m, and resonance frequencies of 200–400 kHz.

The height images have been processed using the software NanoScope Analysis 1.40 (Bruker Corporation, Billerica, MA, USA) and elaborated via OriginPro software (OriginLab Corporation, Northampton, MA, USA). The elaboration of the height profile has been done in accordance with literature [37].

For all the samples, the average surface roughness (*R_a_*) and the root mean square roughness parameter (*R_q_*) were evaluated, according to Equations (6) and (7):(6)$$R_{a}=\frac{1}{l_{r}}\int_{0}^{l_{r}}\left|z\left(x\right)\right|*dx$$
(7)$$R_{q}=\sqrt{\frac{1}{l_{r}}\int_{0}^{l_{r}}z{\left(x\right)}^{2}dx}$$
where *l_r_* is the length of the line, *z* is the height and *x* is the position.

### 3.11. Drug Release

Membrane samples were cut in 1 cm diameter and stirred in PBS in an orbital shaker at 300 rpm. Drug release kinetics were monitored by using a Spectrometer UV-2401 PC (Shimadzu, Kyoto, Japan). The tests were performed using rectangular plates with an exposed area of about 3 cm^2^ and 1 cm of the light path.

For AuM1, the phenyl group was tracked in the release medium to monitor drug release, as previously reported in literature [37]. By tracking the absorbance of the target group (phenyl for AuM1, R-N=N-R for Dacarbazine, and (NHC^d^)AuCl) of known quantities of the various active substances, the calibration curves are well described by the Lambert-Beer Equation (8).
(8)A=ε∗c∗l
where *A* is the absorbance, ε is the absorptivity of the substance, *c* is the concentration of the solution, and *l* is the light path length. Table 6 reports the *ε* values for the complexes.

The peak monitored for AuM1 complex is detected around the expected value (252 nm). For AuM1 spectrum, π→π* transitions are caused by phenyl around 254 nm and around 200 nm. The band at λ = 328 nm has been attributed to the transition n→π* transition that could be from R-N=N-R (that has been tracked for Dacarbazine and (NHC^d^)AuCl). As already reported in the literature, the amidic groups bonded, such as imidazole, can shift from 210 nm to over 225 nm [41]. In particular, it is already reported in literature that the second peak of Dacarbazine in UV-Vis region is at 236 nm [42]. However, it is worth noticing that the lower conjugation in (NHC^d^)AuCl leads to a hypsochromic shift (to 228 nm) of the peak. The spectra are reported in Figure 10.

The absorptivities of the active substances are reported in Table 2.

Observing the spectra, π→π* transitions around 254 nm and around 200 nm caused by phenyl are evident for both complexes dissolved in the solution. Since AuM1 shows the presence of two chlorines on the backbone, it is reasonable that Cl atoms attract electronic density given its high electronegativity, decreasing and shifting the intensity of the π→π* transition of the electrons involved in the double bond of the two carbons. For (NHC^d^)AuCl, a shoulder around 266 nm is evident. This transition is probably due to π→π* double bond transition but is shifted to the AuM1 peak because of the chemical surrounding. In literature can be found in very different ranges of the UV-vis spectrum, from 184 nm for ethylene to over 400 nm for molecules such as β-carotene [43] For release profile fitting, a statistical approach was followed by using a modified Weibull model, see Equation (9), which was recently proposed in literature [38]:(9)$$\frac{M_{t}}{M_{\infty }}=\theta *[1-\exp\left(-\frac{1}{A_{1}}*t^{b_{1}}\right)]+\left(1-\theta \right)*\;[1-\exp\left(-\frac{1}{A_{2}}*t^{b_{2}}\right)]$$
where *M* is the substance amount released at a certain time, *M_∞_* is the total substance amount in the sample (evaluated by weighting the sample and knowing the complex percentage in the membrane), *A*_1_, *A*_2_, b_1_, b_2_ and *θ* are kinetic constant parameters, *t* is the time. The first principles Fick model, reported in Equation (10), has been applied to fit the diffusive release curve, according to Siepmann et al. [36,44], considering the system as a matrix drug delivery system.
(10)$$\frac{M_{t}}{M_{\infty }}=1-\frac{8}{\pi^{2}}*\overset{\infty }{\underset{n=0}{\sum }}\frac{\exp\left(-D*{\left(2n+1\right)}^{2}*\pi^{2}*\frac{t}{L^{2}}\right)}{{\left(2n+1\right)}^{2}}$$
where *L* is the thickness of the membrane and the only fitting parameter is *D*, the diffusivity of the system.

The encapsulation efficiency of the electrospinning process for the various membranes has been evaluated by dissolving the sample in chloroform. In this way, a calibration line of the various active substances in chloroform has been built up. The encapsulation efficiency has been defined as follows in Equation (11).
(11)$$Encapsulation\;Efficiency\;\equiv \;\eta =\frac{m_{active\;substance\;in\;the\;membrane}}{m_{theoretical}}*100$$

## 4. Conclusions

In this paper, electrospun systems loaded with highly active antitumoral substances were proposed. PCL electrospun membranes were loaded with commercial dacarbazine and synthetic AuM1, (NHC^d^)AuCl) antitumoral active substances. The possibility of delivering dacarbazine using topical drug delivery was investigated and validated. Furthermore, two highly active substances were synthesized and tested. They proved to be more active than the traditional drug employed for melanoma treatment. The efficacy is retained even after their inclusion in the electrospun membranes. Furthermore, the present study highlights the key role of matrix-filler affinity in designing drug-delivery devices. From the morphology investigated using AFM, it was observed that the roughness parameters for the membranes loaded with affine and non-affine substances are sensitively different. The filler tends to be segregated on the outer part of the fiber in case of low affinity. In contrast, the inclusion is more efficient in case of attractive interactions between filler and matrix, as confirmed by FTIR analysis. In light of this, it is possible to forecast the drug location in the drug delivery system that, consequently, affects the release profile of the system. Dacarbazine-loaded and (NHC^d^)AuCl-loaded systems follow a Fickian profile, with a more sustained release in time. This is due to the more efficient inclusion of the filler inside the nanofiber. In the case of AuM1-systems, a two-step release is reported, with a highly prominent burst in the first stage because of the location of the complex on the fiber surface.

## Figures and Tables

**Figure 1 ijms-24-01507-f001:**
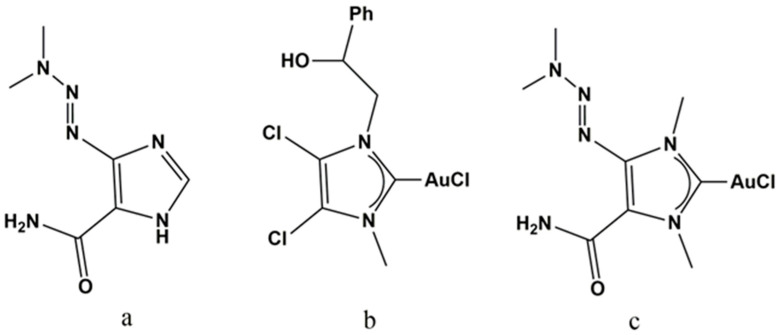
(**a**) Dacarbazine; (**b**) AuM1; (**c**) (NHC^d^)AuCl.

**Figure 2 ijms-24-01507-f002:**
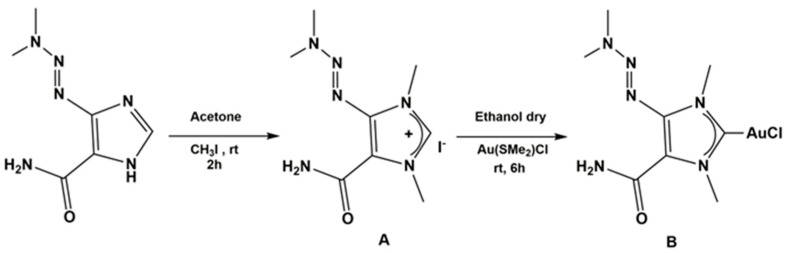
Synthesis of proligand (**A**) and (NHC^d^) AuCl complex (**B**).

**Figure 3 ijms-24-01507-f003:**
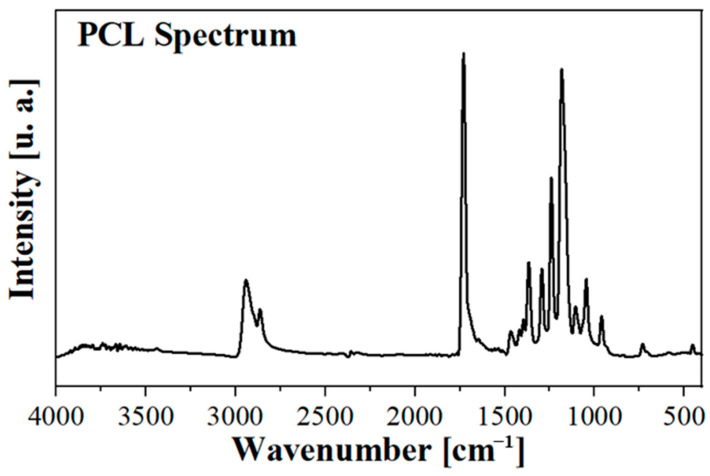
FTIR Spectrum of the PCL membrane.

**Figure 4 ijms-24-01507-f004:**
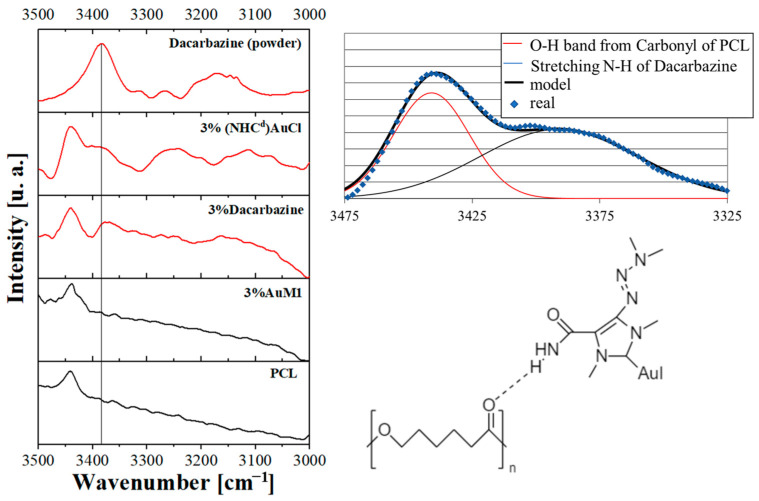
FTIR between 3000–3500 cm^−1^ of Dacarbazine powder and the membranes unloaded (PCL) and loaded with AuM1 (3%AuM1), Dacarbazine (3%Dacar) and (NHC^d^)AuCl (3%(NHC^d^)AuCl).

**Figure 5 ijms-24-01507-f005:**
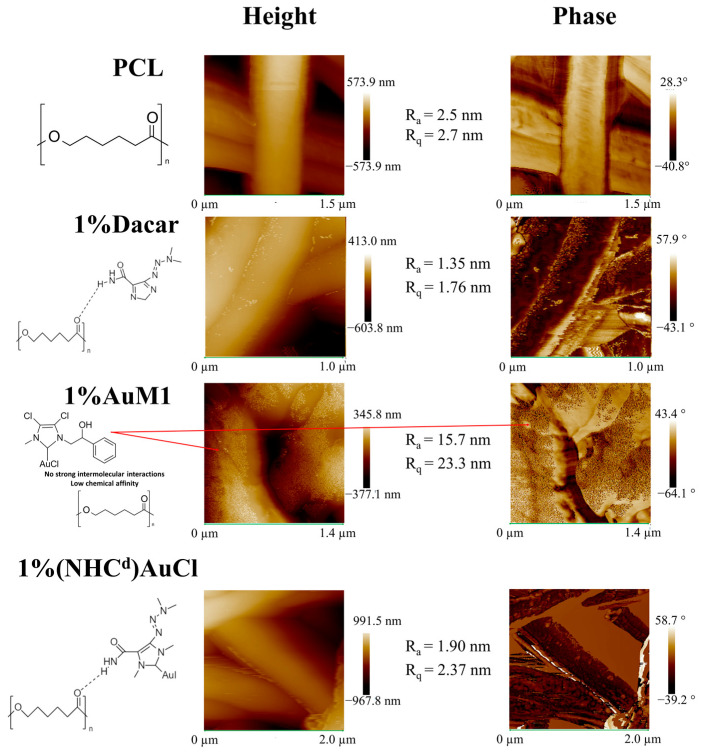
AFM images of the unfilled and filled membranes.

**Figure 6 ijms-24-01507-f006:**
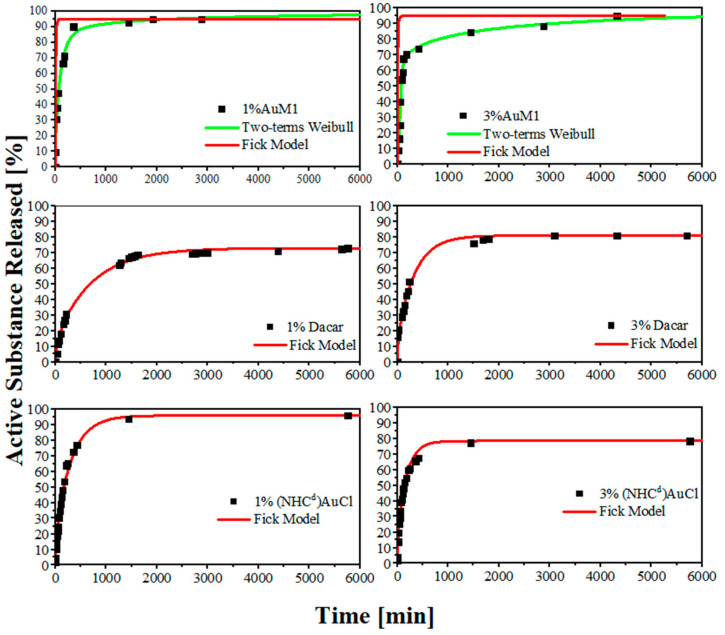
Release curve of the electrospun membranes.

**Figure 7 ijms-24-01507-f007:**
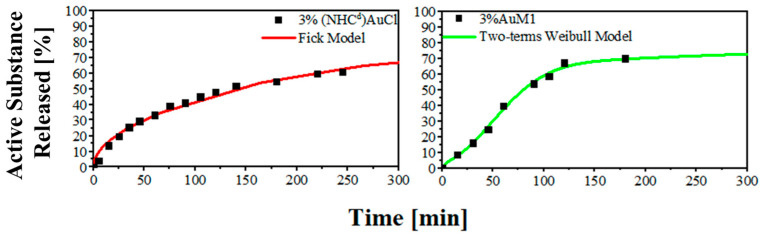
Release kinetic in the first region of the curves.

**Figure 8 ijms-24-01507-f008:**
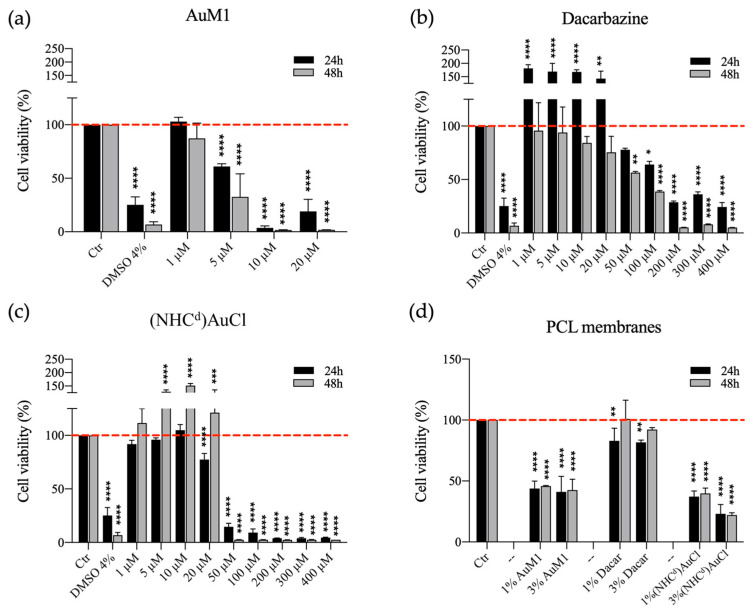
MTT assay on MeWo cells treated with free compounds or seeded on PCL functionalized membranes. 1, 5, 10, 20, 50, 100, 200, 300 and 400 μM of AuM1 (**a**), Dacarbazine (**b**) and (NHC^d^)AuCl (**c**) were tested to evaluate the cytotoxic effect of free compounds on MeWo cells; MeWo viability was also studied seeding cells on PCL membranes (**d**), functionalized with 1% and 3% of each compound. The experiments were analyzed by ordinary one-way ANOVA test. * *p* ≤ 0.05, ** *p* < 0.01, *** *p* < 0.001, and **** *p* ≤ 0.0001 (*n* = 3).

**Figure 9 ijms-24-01507-f009:**
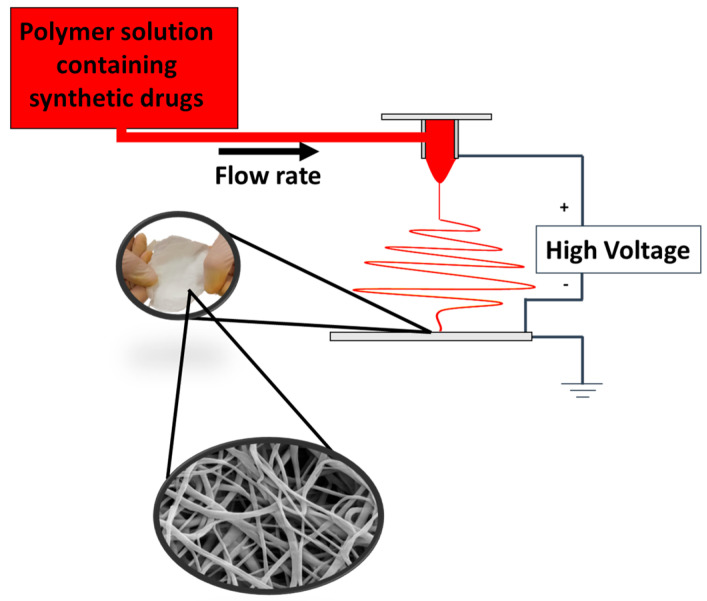
Electrospinning process.

**Figure 10 ijms-24-01507-f010:**
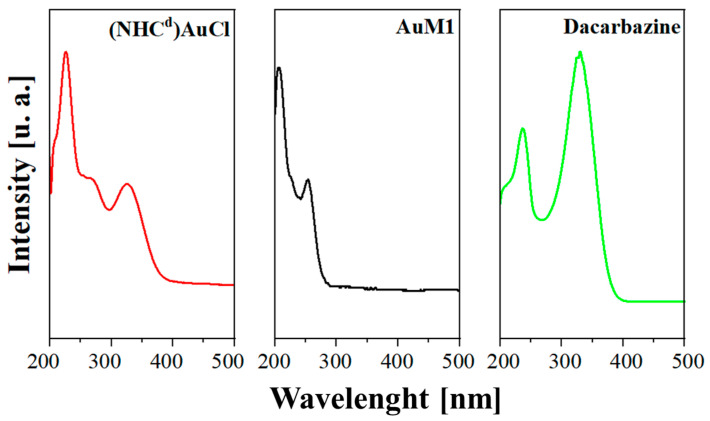
UV-Vis Spectra of Dacarbazine, AuM1 and (NHC^d^)AuCl.

**Table 1 ijms-24-01507-t001:** Roughness parameters evaluated for the various membranes.

#	R_a_ [nm]	R_q_ [nm]
PCL	2.5	2.7
1%AuM1	15.7	23.3
3%AuM1	23.7	28.7
1%Dacar	1.35	1.76
3%Dacar	1.37	1.69
1%(NHC^d^)AuCl	1.90	2.37
3%(NHC^d^)AuCl	4.67	6.21

**Table 2 ijms-24-01507-t002:** Encapsulation efficiency of the systems.

Membrane	η
1%AuM1	98.6%
1%Dacar	98.0%
1%(NHC^d^)AuCl	98.3%
3%AuM1	79.1%
3%Dacar	76.0%
3%(NHC^d^)AuCl	74.8%

**Table 3 ijms-24-01507-t003:** Two-terms Weibull parameters for electrospun systems.

Weibull Parameters	1%AuM1	3%AuM1
θ	0.533	0.544
A_1_ [h^b1^]	1.901	1.429
b_1_ [-]	1.013	2.027
A_2_ [h^b2^]	1.276	4.058
b_2_ [-]	0.284	0.461
R^2^	0.985	0.997

**Table 4 ijms-24-01507-t004:** Diffusive parameters for diffusive electrospun systems.

Electrospun Systems	D [mm^2^/s]	R^2^
1%Dacar	2.34 × 10^−8^	0.992
3%Dacar	4.95 × 10^−8^	0.976
1%(NHC^d^)AuCl	5.44 × 10^−8^	0.980
3%(NHC^d^)AuCl	1.00 × 10^−8^	0.981

**Table 5 ijms-24-01507-t005:** Process Parameters.

#	Flow Rate [mL/h]	Distance Injector-Collector [cm]	Electric Potential Difference [kV]	Active Substance
PCL	2	28.5	21	-
1%AuM1	1	25	24	1% AuM1
3%AuM1	1	25	25	3% AuM1
1%Dacar	1	25	24	1% Dacarbazine
3%Dacar	1	25	24	3% Dacarbazine
1%(NHC^d^)AuCl	1	20	22	1%(NHC^d^)AuCl
3%(NHC^d^)AuCl	0.5	20	16	3% (NHC^d^)AuCl

**Table 6 ijms-24-01507-t006:** Absorptivity of the active substances in PBS solution.

Active Substance	ε [mL × mg^−1^ × cm^−1^]
AuM1 [Peak 252 nm]	9.8847
Dacarbazine [Peak 328 nm]	92.059
(NHC^d^)AuCl [Peak 328 nm]	12.549

## Data Availability

Not applicable.

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
