# Peer review of "Bottom-Up Strategy to Forecast the Drug Location and Release Kinetics in Antitumoral Electrospun Drug Delivery Systems"

_ijms, 2023, doi:10.3390/ijms24021507_

Round 1

Reviewer 1 Report

In the manuscript titled "Bottom-up strategy to forecast the drug location and release kinetics in antitumoral electrospun drug delivery systems", Longo et al. report the synthenthesis of a novel anticancer agent based on Au-complexed NHC compound that is elecgtrospun together with PCL to form drug-loaded membranes for the topical treatment of cancer lesions. The context of this work is well described, as well as the materials and methods used, that allow for fine reproducibility. The results clearly show the effectiveness of the proposed methods and he conclusions are efficiently supported by the experimental results. Therefore, I recommend the publication of this work on the International Journal of Molecular Sciences in its current form.

Reviewer 2 Report

This paper used the electrospun systems to synthesize the polycaprolactone membranes as drug delivery systems and loaded with synthetic drugs. Various characterization methods, including AFM, FTIR etc, have been fully investigated. Finally, the toxic activity towards cells is performed to research antitumoral activity via the MTT method. The whole manuscript looks good, and the writing is well-organized. It can be accepted after minor revision of some issues.

1. A scheme illustration should be given to figure out the synthesis of the drug delivery system via the electrospinning process.

2. It should be compared with other reported research works about the antitumoral activity.

3. Some advanced drug delivery works are missing in the manuscript and should be included, like doi.org/10.1016/j.mattod.2020.02.001, etc. 
